

# Loitering behavior detection by spatiotemporal characteristics quantification based on the dynamic features of Automatic Identification System (AIS) messages

Wayan Mahardhika Wijaya[1] and Yasuhiro Nakamura[2]

[1] Graduate School of Science and Engineering, National Defense Academy of Japan, Yokosuka, Kanagawa, Japan
[2] Computer Science, National Defense Academy of Japan, Yokosuka, Kanagawa, Japan

Corresponding author
Yasuhiro Nakamura, yas@nda.ac.jp

## ABSTRACT

The capability of the Automatic Identification System (AIS) to provide real-time worldwide coverage of ship tracks has made it possible for maritime authorities to utilize AIS as a means of surveillance to identify anomalies. Anomaly detection in maritime traffic is crucial as anomalous behavior may be a sign of either emergencies or illegal activities. Anomalous ships are recognized based on their behavior by manual examination. Such work requires extensive effort, especially for nationwide surveillance. To deal with this, researchers proposed computational methods to analyze vessel behavior. However, most approaches are region-dependent and require a profile of normality to detect anomalies, and amongst the six types of anomaly, loitering is the least explored. Loitering is not necessarily anomalous behavior as it is common for certain types of ships, such as pilot boats and research vessels. However, tankers and cargo ships normally do not engage in loitering. Based on 12-month manually examined data, nearly 60% of the identified anomalies were loitering, particularly for those of types cargo and tanker. Although manual identification is inefficient, automatically identifying abnormal vessels by merely implementing computing algorithms is not yet feasible. It still needs subject matter experts' assessments. This study proposes a region-independent method to automatically detect loitering without training normal instances and produces a ranked list of loitering vessels to facilitate further anomaly investigation. First, the loitering spatiotemporal characteristics are defined: (1) movement of frequent course change, with a certain speed, within a certain spatial range, (2) movement of frequent course change within traversed geodetic distance, (3) might demonstrate frequent extreme turning, and (4) extreme turning produces a significant discrepancy between the course over ground and the heading of the ship. Then, the characteristics are quantified by manipulating the dynamic information of AIS messages. Finally, the parameters to determine a loitering trajectory are formulated by comparing the rate of course change, speed, and the discrepancy between heading and course with the area of spatial range enclosing the trajectory and the geodetic distance between the start and end point. The loitering score of each trajectory is calculated with the parameters, and the Isolation Forest algorithm is employed to establish a threshold and rank. Then, geographic visualization is created for intuitive evaluation. An experiment was conducted on a real-world dataset covering a sea area of 610,116.37

km2. The results prove the efficacy of the proposed method. It remarkably outperforms the existing approach with 97% accuracy and 92% F-score. The experiment produces a ranked list of loitering vessels and an intuitive visualization in the relevant geographic area. In the realworld scenario, they are practical means to support further examination by human operators.

## INTRODUCTION

The worldwide coverage of near real-time vessel movement information has been made available by the widespread application of the shipborne Automatic Identification System (AIS). It was developed in the 1990s for the main purposes of preventing collisions and improving navigational safety (*Yang et al., 2019*). Ships involved in international voyages with a gross tonnage (GT) of 300 or more, cargo ships not engaged in international voyages of 500 GT or more, and all passenger ships, regardless of size are required to be fitted with an AIS device as regulated by the International Convention for the Safety of Life at Sea (SOLAS) 1974 Chapter V, Regulation 19.2 (*IMO, 2019*; *IMO, 2015*; *IMO, 2006*; *IALA, 2004*). According to the world merchant fleet statistics published by the Electronic Quality Shipping Information System (Equasis) in 2021, there are a total of 118,928 merchant ships in the world, of which 53.4% are ships of 500 GT or more and 46.6% are less than 500 GT (*Equasis, 2021*). Vessels of types cargo and tanker account for 74.2% of ships greater than or equal to 500 GT. In other words, there should be more than 47,175 tankers and cargo ships equipped with AIS.

AIS transceives messages containing navigation data among vessels, terrestrial base stations, and/or satellites. The information broadcasted by an AIS transceiver consists of the following: (1) static information (name, call sign, IMO number, MMSI number, dimension, type of ship), (2) dynamic information(position, position timestamp in UTC, speed over ground, course over ground, heading, rate of turn, navigational status), and (3) voyage-related information (destination, estimated time of arrival, draught) (*IMO, 2015*). The transmission interval varies from 2 to 10 seconds, proportional to the speed for ships moving faster than 3 knots, and it is 3 minutes for ships at anchor or moored and not moving faster than 3 knots (*IMO, 2015*).

The capability of AIS to provide real-time information on ships' movement has made it possible for national maritime authorities around the world to utilize AIS as a monitoring tool over their respective sea area of interest. It enables them to conduct comprehensive surveillance of vessel behavior over a nationwide area. Based on their behavior, anomalous ships are identified by manual examination of knowledgeable experienced officers. The detection of anomalous vessels in the maritime domain is particularly important for two main reasons: (1) anomalous behavior could be caused by a mechanical failure of the

vessel and may therefore require rescue at sea, and (2) anomalous behavior can serve as an indirect measure of illegal actions (*Kullberg, Skog & Hendeby, 2021*).

However, in the case of nationwide surveillance that may involve thousands of ships, the manual work is inefficient as they require extensive effort and time. Furthermore, it may cause oversight and leave anomalous vessels undetected. To deal with the hurdle, researchers proposed computational methods to analyze vessel behavior for anomaly detection and movement prediction (*Tu et al., 2018*). Those researches are facilitated by the openly accessible high-quality historical AIS data such as the AIS data of MarineCadastre.gov that is used in the experiment of this study.

In the case of anomalous behavior detection, *Wolsing et al. (2022)* conducted an extensive review of 44 publications that specifically aimed at anomaly detection in AIS tracks. The papers were published within the range years 2007 to 2021. They found that most of the research, 38 out of 44, is region-centric, which is heavily dependent on historical AIS data of the region. They require re-training if the application shall be implemented in other regions. In addition, amongst the five types of anomalous ship behavior defined by *Lane et al. (2010)*, which is referenced by the review, 37 out of 44 publications examine deviation from standard routes anomaly, seven approaches consider unexpected AIS activity, and none of the papers discussed loitering behavior, a type of maritime anomalies mentioned by *Laere & Nilsson (2009)*.

Loitering is not necessarily anomalous behavior as it is common for certain types of vessels, such as recreational crafts, Search and Rescue (SAR), research, law enforcement, military, fishing, and navigation-supporting vessels (*Zhang et al., 2022*). However, based on the nature of their operations and the requirement to comply with the guideline of efficient and safe navigation (*IMO, 2000*), tankers and cargo ships typically do not engage in loitering as part of their regular operations. According to the 12-month ship anomaly data provided by the Indonesian Coast Guard, selected and examined manually by experts, roughly 97% of the anomalous vessels are of types cargo and tanker, and nearly 60% of the identified anomalies were loitering vessels.

Another tendency in maritime anomaly detection research is that the proposed algorithms involve building profiles of normal instances for traffic patterns based on historical movement data and applying the models to identify anomalous vessels (*Wolsing et al., 2022*; *Tu et al., 2018*).

Given the above findings, advancement in the subject of maritime anomaly detection can be carried out by untangling the remaining limitations: (1) region dependency, (2) the need for training normal instances, (3) loitering anomaly has not been sufficiently examined, and (4) manual selection and examination of anomalous vessels.

Although manual identification is inefficient, automatically identifying anomalous vessels by merely implementing computing algorithms is not yet feasible since it still needs subject matter experts' consideration based on knowledge, years of experience, and timely assessment of the ongoing context. Computational techniques always need a threshold to delineate anomaly from normal. Meanwhile, in the dynamic of ship behavior in real-world situations, anomaly and normal events often overlap with each other. Vessels displaying

abnormal behavior do not automatically indicate their definite involvement in illegal activities (*Ford et al., 2018*).

Motivated by the remaining issues, this study is aimed at developing a region-independent method to automatically detect loitering behavior without the need to train normal instances. The method defines the spatiotemporal characteristics of loitering behavior. The characteristics are quantified with the rate of course change, speed, the discrepancy between heading and course, the area of spatial range enclosing the trajectory, and the geodetic distance between the starting and ending point of the trajectory. Then, the parameters to determine a loitering trajectory are formulated by comparing the rate of course change, speed, and the discrepancy between heading and course with the area of spatial range enclosing the trajectory and the geodetic distance between the starting and ending point. The loitering detection is performed by applying the formulated parameters combined with the employment of the Isolation Forest algorithm to produce a list of loitering vessels ranked with the score of anomaly. Additionally, the vessels' trajectories are geographically visualized for intuitive recognition and evaluation. In the real-world scenario, the ranked list and the geographic visualization are practical means to facilitate an efficient and comprehensive examination by human operators.

The proposed approach in this article is expected to contribute in the following ways: (1) Discover a versatile loitering detection method that does not require a profile of normal instances, and runs on decent commodity servers. It does not involve special hardware requirements such as GPU. (2) Provide practical solutions for authorities to automatically detect unexpected behavior of vessels in dense fairways or harbors approaching areas. (3) Produce a ranked list of loitering vessels and intuitive geographic visualization of their trajectories that enable human operators to work efficiently and reduce oversight. The remainder of this article is organized as follows. The next section reviews existing research relevant to this study. Following this, the 'Materials and Methods' section defines the study area and describes the method for detecting loitering behavior in maritime trajectories. Then, the experimental results, analysis, and evaluation are presented in the Results and Discussion section. Finally, the last section concludes the content of this article and specifies future works.

## LITERATURE REVIEW

An anomalous behavior is defined as a behavior that is inconsistent with or deviating from what is usual, normal, or expected, or that is not conforming to rules, laws, or customs (*Laere & Nilsson, 2009*). Within the range of behaviors that can be monitored using AIS, five types of anomalous ship behaviors were proposed by *Lane et al. (2010)*: (1) deviation from standard routes, (2) unexpected AIS activity (AIS on/off), (3) unexpected port arrival (ship of a particular type arrives at a port that has no appropriate facilities to handle the ship), (4) close approach (two ships being unusually close together at sea and traveling below a certain speed), and (5) zone entry (ship navigates to a restricted zone such as military installations, national infrastructure, protected environment, or area of bad weather). These five types of anomalous behaviors classification are adopted by *Wolsing et*

*al. (2022)* in their review of 44 recent approaches specifically designed for anomaly detection in AIS tracks. They found out that most studies, 37 out of 44, explored the deviation from standard or well-known shipping routes by focusing on confined regions rather than training universally applicable models. This type of approach is heavily dependent on the AIS tracks of the specifically defined region. Thus, retraining and re-evaluation of new models are needed when it is to be applied to a new region of the sea. Wolsing et al. concluded that with the increasing availability of AIS datasets and the fact that vessels navigate on predictable routes, automated anomaly detection is a plausible approach to detect unintended or even malicious behaviors.

In the case of deviation from standard routes anomaly, Lamm and Hahn presented a normalcy model of ship maneuver by generating Maneuver Net from historical AIS data to detect unplanned maneuvers, especially in restricted waterways (*Lamm & Hahn, 2019*). The model implements cumulative sum (CUSUM) procedure (*Lamm & Hahn, 2017*) and DBSCAN algorithms (*Ester et al., 1996*) to detect maneuvers and for clustering the maneuvers respectively. Another approach is presented by *Wang et al. (2020)* which implemented kNN-based clustering to detect rare behavior in maritime trajectories in a visual analytics manner. The work assumpts that both useful and useless anomalies co-exist within the massive AIS database. Only extremely few anomalies are useful, they are rare behaviors. The whole process of the method involves trajectory reconstruction, an association of each vessel to a sampled trajectory, high dimensional trajectory data processing, k-NN search in a GPU-accelerated in-memory database, and the implementation of Local Outlier Factor (LOF) algorithm for post-verification. Finally, it provides the operators in the surveillance office with a rare behavior factor (RBF) to adjust the detection sensitivity based on their knowledge of the circumstances of the target area.

In addition to the five types of anomaly, *Tu et al. (2018)* took a different perspective and classify ship anomaly as: (1) position anomaly, which is if a ship appears in a restricted/forbidden region or in an unexpected position, (2) speed anomaly, that is when the speed of a ship is significantly above or below regular speed in the same context, and (3) time anomaly, when the visiting time of a ship is unexpected. To detect these anomalies, detection algorithms are designed to construct models of normalcy for traffic patterns from historical motion data and utilize the models to identify vessels with anomalous characteristics. These algorithms generally implement statistical and/or machine-learning methodologies. Laxhammar implemented the Gaussian Mixture Model (GMM) to model normal sea traffic patterns and use the learned model to detect anomalies in sea traffic (*Laxhammar, 2008*). The proposed method is proven effective in detecting rather elementary-type anomalies such as speeding, crossing sea lanes, and traveling in the wrong direction.

A more recent approach in abnormal vessel detection based on AIS tracks was proposed by *Shi et al. (2022)* which analyzed the spatial position and thematic attributes abnormalities. The spatial position refers to the position of the ship in longitude and latitude coordinates, while the thematic attributes consist of speed, course over ground (COG), average speed, and average acceleration. The method extracts maritime routes

based on vessel trajectories derived from AIS tracks. A ship's spatial position is examined with the extracted routes, then Rayda's criterion is applied to evaluate the abnormality of the ship. Rayda's criterion takes k ($k = 1, 2, 3$) times the standard deviation as the threshold value. The Isolation Forest algorithm (*Liu, Ting & Zhou, 2008*) is applied to the thematic attributes of the ship, and the abnormality is determined with the score resulting from the algorithm. The experimental results show the effectiveness of the method. However, since it detects abnormality based on the ship's movement features (position, speed, course, *etc.*) on a certain point of the ship's trajectory, the method can not be applied to detect loitering behavior. A loitering behavior is identified by examining the ship's trajectory within a certain time duration, it can not be detected by analyzing just one point of the trajectory.

Loitering in maritime traffic has already been specified as a type of anomaly in 2009 (*Laere & Nilsson, 2009*). However, none of the related works discussed in the previous paragraphs tackle loitering behavior. In 2018, a paper by *Filipiak et al. (2019)* described seven subcategories of an anomaly in AIS tracks that is related to loitering behavior: (1) invalid coordinates, (2) invalid location, (3) invalid speed, (4) sharp course change, (5) unpredicted location, (6) speed unusually high, and (7) speed unusually low. The paper uses these subcategories as parameters to detect loitering. However, it does not provide any accuracy or precision evaluation to prove the effectiveness of the method.

A recent publication on loitering behavior detection is presented by *Zhang et al. (2022)* motivated by the scarcity of effort to explore motions and behaviors characteristics of ships' frequent turning within a certain range, a.k.a. loitering. The importance of the research lies in the assumption that a ship loitering in a certain area implies its deep interest in the place or something inside. The paper introduces the concept of trajectory redundancy (TR) to detect loitering trajectories from AIS tracks. The calculation formula for TR is Eq. (1), where TR is represented by $\psi$, $D$ is the length of ship trajectory, and $P$ is the perimeter of the minimum bounding rectangle of ship trajectory. The larger $\psi$ is the greater the possibility of loitering behavior. The threshold is $\psi_{min} = 1$. Figure 1 shows three different trajectories in the same size of the spatial range (all have the same $P$) with $\psi < 1$, $\psi \approx 1$, and $\psi > 1$.

$$\psi = D/P \tag{1}$$

However, since TR does not consider the movement speed along the trajectory, ships that travel on exactly the same trajectories within the same duration will produce the same TR value even though their speed is different. For example, a ship maneuvering on one perfect circle in slow motion for 24 h within a harbor will have the same TR value as the one traveling with high speed for 24 h on the same shape of trajectory over the Pacific Ocean. The illustration is as Fig. 2. If the speed ($S$) of the ships is included in the TR calculation as in Eq. (2), then each ship's TR will yield a different value. Here, speed refers to the speed over ground (SOG).

$$\psi = \frac{D}{P \times S} \tag{2}$$

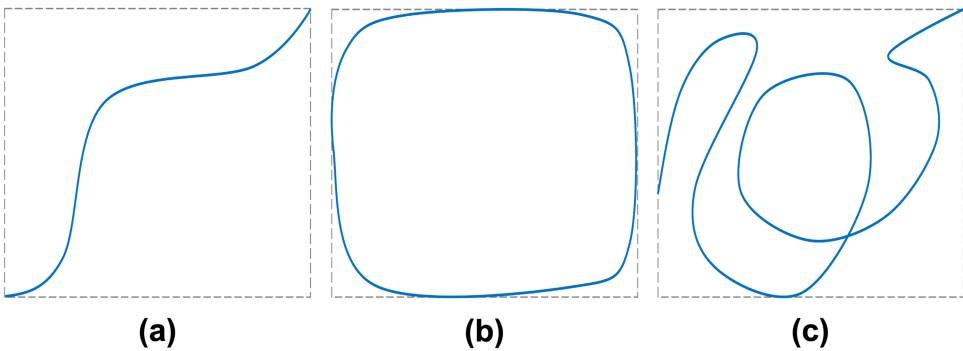

**Figure 1** **Three different trajectories with the equivalent TR ($\psi$) values.** The grey dashed lines represent the minimum bounding rectangle of each trajectory, while the blue solid lines depict vessel trajectories. (A, B, C) are the trajectory with $\psi < 1$, $\psi \approx 1$, and $\psi > 1$ respectively.

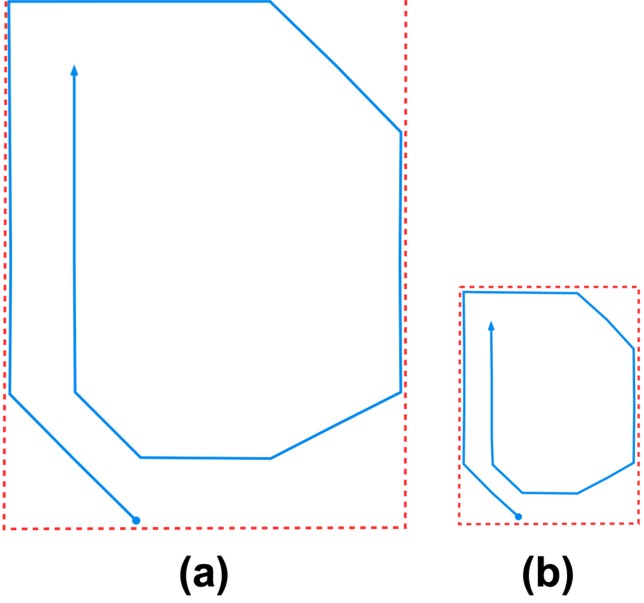

**Figure 2** **The blue lines and red dashed line depict ship trajectories and its bounding rectangles respectively.** Within the same duration, both (A, B) have the same TR value as both their trajectory length and the bounding rectangle perimeter are proportional. If speed parameter is included as in Eq. (2), then the difference will become obvious.

Furthermore, as TR does not include the change of course parameter in the calculation, ships that travel on different trajectories may yield the same TR value as depicted in Fig. 3. If the change of course ($\Delta C$) is included in TR calculation as in Eq. (3), then the difference amongst the trajectories may be revealed. Thus, this article proposes an extension to the TR by including speed and change of course features as in Eq. (8) described in the 'Materials and Methods' section.

$$\psi = \frac{D \times \Delta C}{P} \tag{3}$$

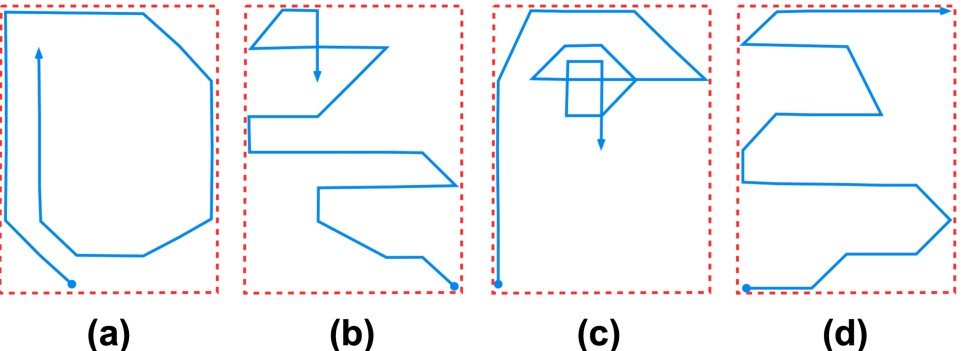

(a)         (b)         (c)         (d)

**Figure 3** **These four different trajectories: (A, B, C and D) have the same TR values as they have the same length within the same perimeter of the bounding rectangle.** The blue lines and red dashed lines depict ship trajectories and its bounding rectangles respectively. If the change of course parameter is included as in Eq. (3), then the difference may be revealed.

Derived from the variety of the detected loitering trajectories, *Zhang et al. (2022)* propose four types of loitering shapes as illustrated in Fig. 4: (1) disordered retracting shape, (2) lasso shape, (3) regular reciprocating shape, and (4) random coil shape. Further, it uses these loitering shapes as a model to classify loitering trajectories of real-world AIS tracks. The convolutional neural networks (CNN) algorithm is applied to learn the shape features acquired from the computer vision perspective. Here, each trajectory is processed as a gray image. The experimental results proved that the proposed technique is effective in detecting and classifying loitering behavior. Additionally, based on the type of ship involved, it is also capable of specifying the possible intentions of loitering ships: (1) recreational, (2) fishing, (3) navigation-supporting, and (4) special loitering activities (law enforcement, military operation, search and rescue, research). However, due to the plain definition of loitering behavior expressed by trajectory redundancy (TR) as of Eq. (1), there should exist a space for improvement by a more sophisticated approach. In addition, the article does not discuss loitering behavior as an anomaly in maritime traffic, and it does not involve vessels of types cargo and tanker. Although their method has been proven effective in detecting the types of vessels that typically engage in loitering behavior, its accuracy has not yet been verified for cargo and tanker vessels, the types that do not normally loiter.

In this article, following the Guideline of Voyage Planning mandated by the International Maritime Organization (*IMO, 2000*; *Skóra & Wolski, 2016*) and considering the works of *Laere & Nilsson (2009)* and of *Filipiak et al. (2019)*, loitering behavior is adopted as a type of anomalous behavior in the context of vessel movement. According to the guideline, the voyage of a ship is a deliberately planned event that should assure the safety of life at sea, efficient and safe navigation, and protection of the marine environment. Thus, in normal circumstances, any ship shall not take any maneuver that endangers people's life at sea, they shall navigate as efficiently as possible in a safe manner, which means that they are to take the shortest and fastest route whenever it is safe to do so, and they shall not conduct

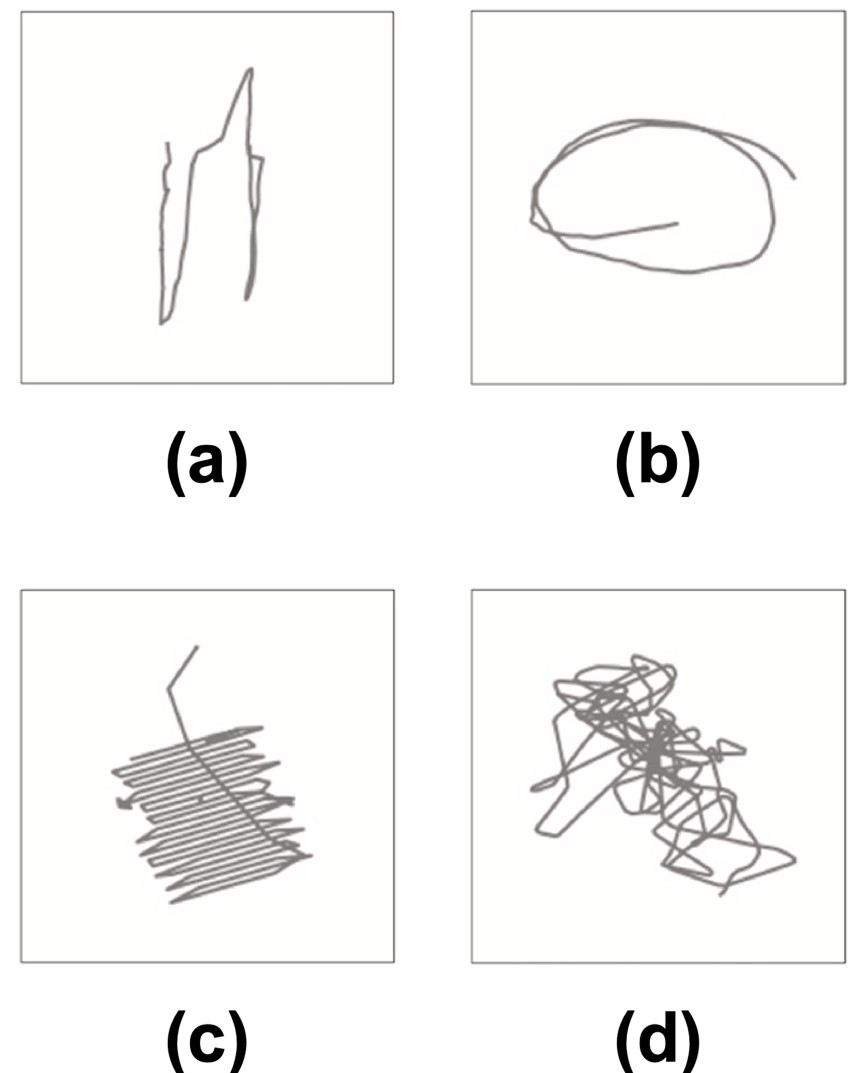

**Figure 4** Four shapes of loitering trajectory: (A) disordered retracting shape, (B) lasso resembling shape, (C) reciprocating shape, and (D) random coil shape.

any activity that causes pollution or damage to the marine environment. Any ships that behave oppositely are to be considered anomalous.

Although state-of-the-art methods of anomaly detection based on AIS tracks demonstrate effectiveness in detecting anomalous vessels, determining an optimal threshold to delineate anomaly from normal often requires hyperparameters fine tuning, *e.g.*, the *eps* and *MinPts* parameters of the DBSCAN (*Ester et al., 1996*) and the *contamination* parameter of the Isolation Forest algorithm (*Liu, Ting & Zhou, 2008*; *Luan et al., 2021*). Another approach such as in the work of *Shi et al. (2022)* Rayda's criterion is applied which takes $k$ times the standard deviation as the threshold value, where $k = 1, 2, 3$. When the selected threshold does not match the operators' perception of abnormality, there will be a discrepancy between the algorithm results and the operator's expectation. Since

the perception of anomaly is dynamic, evolves over time, and the judgment often needs expert's knowledge accumulated over years of study and experience as well as proportional consideration of the relevant circumstances, even the most advanced method is not yet capable of producing the ideal solution. For example, an anomalous ship detected of unexpected activity (AIS on/off) may not be an anomaly in the real scenario. Even though AIS can be switched off to cover illicit operations, the absence of AIS messages might be just caused by a momentary loss of signal. To such a case, *Mazzarella et al. (2017)* proposed a technique for detecting whether a shortage of AIS messages represents an alerting situation or not, by exploiting the Received Signal Strength Indicator available at the AIS Base Stations. The algorithm was applied to real-world data and was able to make the false alarm probability fall below 10% while keeping the detection probability above 90%. However, the evaluation was limited to a single AIS Base Station, thus it is not relevant for a surveillance center scenario that integrates AIS messages from multiple sources to cover a nationwide area.

According to the discussed previous works, this article adopts six types of anomalous ship behaviors that can be observed on AIS tracks: (1) deviation from standard routes, (2) AIS on/off, (3) arrival of ship at inappropriate port, (4) close approach, (5) zone entry, and (6) loitering, of which is the least explored. Furthermore, the discussion identified multiples issues remaining in the previously proposed methods: (1) the majority of the approaches are so heavily dependent on AIS data of the designated region that re-training is required should it be applied in other sea areas, (2) there is a tendency to rely on the training of normal instances as most detection algorithms build normalcy profiles of vessel behavior patterns and apply the models to identify anomalous vessels, (3) several methods involve extensive computation, (4) loitering behavior is rarely explored, and many methods are not applicable for loitering detection as anomalous characteristics are measured on a point basis instead of on segment of trajectory, (5) the existing method of loitering detection does not provide comprehensive features of loitering behavior, does not discuss vessel of type cargo and tanker, and does not examine loitering as an anomaly in maritime traffic, (6) advanced work on detecting intentional AIS on/off is not relevant to a nationwide area of the surveillance system. This article examines the underlying characteristics of vessel loitering behavior, recognizes loitering as a type of anomalous behavior, especially for ships of types cargo and tanker, and develops a loitering detection method that overcomes the remaining issues of the known methods.

## MATERIALS AND METHODS

In the context of vessel movement, loitering was discussed as an anomalous behavior in the works of *Laere & Nilsson (2009)* and of *Filipiak et al. (2019)*. However, the papers do not specify the definition of loitering, specifically in the context of vessel behavior. The work of *Zhang et al. (2022)* defines loitering as a behavior of frequent turning within a certain spatial range. Proceeding from the definition, loitering detection parameters are formulated as in Eq. (1).

However, vessels at anchor, which have their engine stopped, also experience frequent turning or change of course within a certain spatial range due to the effect of current and

wind. Thus, in this article, the definition of loitering behavior is extended to be a behavior of frequent turning within a certain spatial range with a certain speed. If the speed feature is not considered, ships at anchor could be misidentified as loitering. Stem from the definition, the spatiotemporal characteristics of loitering can be elaborated as follow: (1) movement of frequent course change, with a certain speed, within a certain spatial range, (2) movement of frequent course change within traversed geodetic distance, (3) might demonstrate frequent extreme turning, and (4) extreme turning produces a significant discrepancy between the course over ground (COG) and the heading of the ship. These spatiotemporal characteristics of the loitering behavior correspond to the dynamic information of the AIS messages. Course change is the absolute difference between COG at two consecutive points. Speed is the speed over ground (SOG). The spatial range is the bounding box that encloses the ship's trajectory. Geodesic distance is the shortest distance between the starting point and the ending point of the trajectory. The discrepancy between COG and heading is the absolute value of *COG–Heading*.

## Using the dynamic information of AIS messages to quantify the spatiotemporal characteristics of loitering behavior

In order to quantify the spatiotemporal characteristics of loitering behavior, this article utilizes the dynamic information of AIS messages: course over ground (COG), speed over ground (SOG), position, heading, and timestamp (basedatetime). COG is the actual direction of a vessel's movement between two points and SOG is the actual speed, with respect to the surface of the earth. COG is expressed in 360° angular direction, where $0° \leq COG < 360°$. SOG is in knots, which is nautical miles (Nm) per hour. The heading is the direction of the ship's bow as indicated by the compass of the ship, which is also expressed in 360° angular direction within the range of $0° \leq Heading < 360°$. Ship position is expressed in longitude and latitude coordinates, thus ship position at a timestamp can be specified as $point(x, y)$ where $x$ is the longitude and $y$ is the latitude. AIS devices broadcast this dynamic information as a message timestamped in UTC. The broadcasts are transmitted discretely on a certain time interval proportional to the speed of the ship.

Let $M$ be the messages transmitted by a shipborne AIS, Table 1 shows the partial data included in every transmitted message $m$, for $m \in M$. MMSI stands for Maritime Mobile Service Identity, is a unique nine-digit number used by maritime digital selective calling (DSC), automatic identification systems (AIS) and certain other equipment to uniquely identify a ship or a coast radio station (*FCC, 2022*). Similar to a cell phone number, an MMSI number is a unique calling number for DSC radios or an AIS unit (*BoatUS, 2023*). BaseDateTime is the timestamp in UTC. A trajectory of a ship $T$ within a certain spatial range can be constructed by a set of $M$ sorted by timestamp from the starting timestamp $t_0$ to the ending timestamp $t_n$. It can be expressed as $T = \{m_{t0}, m_{t1}, m_{t3} \dots m_{tn}\}$. The area of the spatial range $B$ of a trajectory is the area of the bounding box of the trajectory as depicted by Fig. 5. The geodesic distance $G$ is the shortest distance between the starting point $m_{t0}$ and the ending point $m_{tn}$. This article calculates the geodesic distance $G$ with GeoPy (*Dubrava, 2018*), a Python package that implements the geodesics algorithms proposed by *Karney (2013)*. Finally, the discrepancy between COG and heading $\Delta H$ is calculated as Eq. (4),

**Table 1  Sample of data transmitted in AIS messages from the starting timestamp $t_0$ to the ending timestamp $t_n$.**

|          | MMSI      | BDT[1]     | Lat[2] | Lon[3]    | SOG  | COG   | Heading |
|----------|-----------|------------|--------|-----------|------|-------|---------|
| $m_{t0}$ | 211311970 | 1625316226 | 39.999 | −124.435  | 18.4 | 160.1 | 162     |
| $m_{t1}$ | 211311970 | 1625316291 | 39.994 | −124.432  | 18.5 | 159.2 | 162     |
| …        | …         | …          | …      | …         | …    | …     | …       |
| $m_{tn}$ | 211311970 | 1625324356 | 39.363 | −124.149  | 17.7 | 162.3 | 163     |

**Notes.**
[1] BDT for BaseDateTime.
[2] Lat for Latitude.
[3] Lon for Longitude.

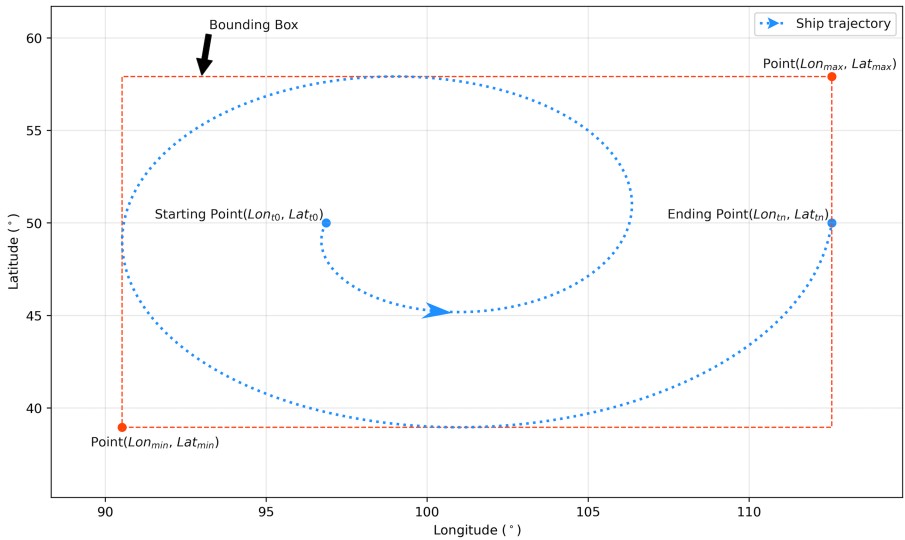

**Figure 5  Illustration of the bounding box of a ship trajectory.** The spatial range $B$ and the perimeter $P$ of a ship trajectory are the area and the perimeter of its bounding box respectively.

and speed $S$ refers to the SOG so that $S_k = SOG_{tk}$, where $0 \leq k \leq n$.

$$\Delta H_k = |COG_{t_k} - Heading_{t_k}| \tag{4}$$

Change of course, denoted $\Delta C$, is the absolute value between COG of two consecutive points as in Eq. (5) for $0 < k \leq n$. Considering the interval time of AIS messages transmission, which is 2 to 10 seconds proportional to speed for ships moving faster than 3 knots, 180° is the largest realistic change of course, especially for the bigger types of vessels such as cargo and tanker. Thus, $\Delta C$ is conditionally calculated as in Algorithm 1.

$$\Delta C_k = |COG_{t_k} - COG_{t_{(k-1)}}| \tag{5}$$

## Construction of comprehensive detection parameters

The frequent change of courses in a loitering trajectory may include minor ones to the relatively large change of courses that occur in extreme turnings. The rate of course change

---

**Algorithm 1** Conditional calculation of $\Delta C$

---

**Require:** : $k > 0$

$\quad \Delta C_k \leftarrow abs(COG_{t_k} - COG_{t_{(k-1)}})$

$\quad$ **if** $\Delta C_k > 180$ **then**

$\quad\quad$ return $360 - \Delta C_k$

$\quad$ **else**

$\quad\quad$ return $\Delta C_k$

$\quad$ **end if**

---

can be described by comparing $\Delta C$ with the maximum course change ($180°$). Considering the area of the bounding box $B$ enclosing the loitering trajectory, the speed $S$ of the ship, and the rate of course change, the score of loitering $F(c)$ can be expressed as Eq. (6). The unit for $B$ and $S$ are $Nm^2$ and *knots* respectively. Here, the score of loitering $F(c)$ is proportional to the rate of course change and inversely proportional to the area of the enclosing bounding box.

$$F(c) = \frac{\sum_{k=1}^{n} \Delta C_k \times \sum_{k=0}^{n} S_k}{180 \times B} \tag{6}$$

Another approach to describe loitering behavior is to additionally consider the discrepancy between COG and heading ($\Delta H$) and the geodesic distance ($G$) between the starting and ending points of the trajectory. In other words, it is to take into account all of the loitering behavior spatiotemporal characteristics as expressed by Eq. (7). The unit for $G$ is Nautical miles ($Nm$).

$$F(c, h, d) = \frac{\sum_{k=1}^{n} \Delta C_k \times \sum_{k=0}^{n} \Delta H_k \times \sum_{k=0}^{n} S_k}{B \times G} \tag{7}$$

The effectiveness of these loitering detection parameters is evaluated in the Results and Discussion section. For this purpose, these parameters are compared with the trajectory redundancy (TR) proposed by *Zhang et al. (2022)* as Eq. (1) and the extension of TR discussed in the next Subsection as Eq. (8).

## Extension of the previous detection parameter

In 'Literature Review' section, the drawbacks of the trajectory redundancy (TR) as a loitering detection parameter as Eq. (1) have been discussed. The possible improvements of the TR by involving speed $S$ as Eq. (2) and change of courses $\Delta C$ as Eq. (3) have also been proposed. This article integrates Eqs. (2) and (3) to formulate an extension of Eq. (1) as a more comprehensive loitering detection parameters. The integrated formula is expressed by Eq. (8). The length of trajectory is calculated as the sum of the geodesic distance $d$ between two consecutive points as Eq. (9) where $0 < k \leq n$. And, $P$ is the perimeter of the trajectory bounding box as illustrated in Fig. 5.

$$\psi = \frac{\sum_{k=1}^{n} d_k \times \sum_{k=1}^{n} \Delta C_k}{P \times \sum_{k=0}^{n} S_k} \tag{8}$$

$$d_k = GeodesicDistance(m_{t_{(k-1)}}, m_{t_k}) \tag{9}$$

## Loitering behavior detection method to produce ranked list of anomalous Vessels

In this article, the detection of loitering behavior is intended to support the unmasking of anomalous vessels hidden in the crowd of maritime traffic. The targets are vessels of types tanker and cargo based on the following reasons:

1. Constrained by strict regulations, economic and safety requirements, tanker and cargo ships should be the most likely to voyage efficiently as well as ensure safety and compliant with the designated standards as mandated in IMO's Guidelines for the Voyage Planning (*IMO, 2000*). Thus, in normal scenarios, vessels of these types shall not loiter.

2. The previous work by *Zhang et al. (2022)* identifies four possible intentions of loitering ships based on the types of vessel involved: recreational for pleasure craft and sailing vessels, fishing for fishing vessels, navigation-supporting for tugboats, and special loitering activities for law enforcement, military operation, SAR (search and rescue), and research vessels. None are of types cargo or tanker, which means they do not have an acceptable reason to loiter.

3. According to the 12-month anomaly ship data of the Indonesian Coast Guard, roughly 97% of the loitering vessels are of types cargo and tanker.

The method for loitering detection is designed to involve the following steps: (1) AIS data preparation and pre-processing, (2) stopping trajectory removal, (3) the application of loitering detection parameters to extract loitering trajectory, (4) implement the Isolation Forest algorithm to produce a list of prospective anomalous vessels ranked with the score of anomaly, and (5) trajectory visualization for an intuitive recognition and evaluation.

The method proposed in this article is developed upon an assumption that maritime traffic surveillance is administered on a certain geographic area of the sea and AIS messages are collected from the vessels navigating through the area. Since this study performs analytics on vessel trajectories, AIS messages received from vessels of the specified geographic area need to be recorded for a certain duration to allow the construction of trajectories. In AIS data preparation and pre-processing, first of all, the recording duration of AIS messages is specified, *e.g.*, 1 day, 3 days, and 5 days. Then, the recorded AIS messages, as depicted in Table 1, are filtered to remove messages with invalid MMSI number: $length(MMSI) \neq 9$, and to select messages from vessels of types cargo and tanker. AIS messages denote vessel types as vessel type codes. The filter $70 \leq VesselType \leq 89$ includes all vessels of types cargo and tanker (*USCG, 2023*). In a busy traffic scenario, the AIS messages are received from multiple vessels. Thus, the method needs to sort the received messages by MMSI and BaseDateTime (timestamp) in ascending mode. Next, the time interval between the timestamps of two consecutive points $\Delta t$ of each vessel is calculated. Any vessel's messages that contain $\Delta t$ bigger than 30 min are to be removed. This is to ensure the time interval between points that construct the trajectory of each ship shall be at least 30 min.

Stopping trajectory removal is intended to remove all trajectories of vessels at anchor, moored vessels, and aground vessels. AIS messages contain information of vessel

navigational status expressed in integer: $0 \leq NavigationalStatus \leq 15$ (*USCG, 2023*).
Removing any messages with navigational status 1, 5, and 6 should remove data of
vessels at anchor, moored vessels, and aground vessels respectively. However, navigational
status is manually set by the vessel's crew (*IMO, 2015*). It is possible that the reported
navigational status is different from the real one due to human error. Thus, removal based
on the bounding box of the anchoring vessel trajectory is also applied. For this purpose,
this article implements a statistical approach by calculating the bounding box area of every
vessel at anchor in the historical AIS data of the relevant sea region. The modified Z-Score
method (*IBM, 2023*) as Eq. (10) is applied to remove outliers of the calculation results.
*MAD* is median absolute deviation and $k = 1.486$. If $M_i > 3$ then $x_i$ is an outlier. After
the removal of the outliers, the maximum value of the remaining instances is used as the
threshold to determine whether a trajectory belongs to an anchoring vessel or not.

$$M_i = \frac{x_i - \tilde{x}}{MAD \times k} \tag{10}$$

The application of loitering detection parameters to extract loitering trajectories is
executed upon completing the calculation of the change of course $\Delta C$, the discrepancy
between COG and heading $\Delta H$, and the geodesic distance $d$ between two consecutive
points for each trajectory. Then, it is required to specify the length of the time window,
*e.g.*, 12 h, 24 h, and 36 h. In the preceding subsection, the length of the trajectory is defined
as the sum of geodesic distance $d$ between two consecutive points from the starting to the
ending point of the trajectory. The length of the time window is equivalent to the length
of the trajectory in the time dimension, which is the length of time between the starting
point $m_{t0}$ and the ending point $m_{tn}$ of the trajectory $T = \{m_{t0}, m_{t1}, m_{t3} \ldots m_{tn}\}$. This time
window is to slide from the starting to the ending BaseDateTime of the AIS messages
recording duration. At the same time, the loitering score is calculated with the loitering
detection parameters (Eqs. (6) and (7)) at every position of the sliding time window. The
scores are compared, and the maximum score is returned as the loitering score of the
respective vessel. Figure 6 illustrates how the time window slides from the starting to the
ending BaseDateTime of the AIS messages recording duration and catches the trajectory
segment with the highest loitering score.

Next, the loitering scores of all processed vessels' trajectories are to be analyzed with
the Isolation Forest algorithm (*Liu, Ting & Zhou, 2008*). The algorithm determines the
threshold of the loitering scores and delineates the loitering trajectories from the normal
ones. The Isolation Forest algorithm provides the capability of detecting anomalies without
the requirement to build a profile of normal instances. Most model-based approaches
construct a such profile and identify instances that do not comply with the normal profile
as anomalies. The idea of the algorithm is that anomalies are 'few and different'. Few means
anomalies are minority in number of instances, and different means anomalies have values
that are significantly different from those of normal ones. The idea corresponds to the
reality that vessels of types cargo and tanker do not normally loiter. Loitering tanker or
cargo ships should be rare, and loitering trajectories is obviously different from the normal
voyage routes that follow the guidelines of voyage planning.

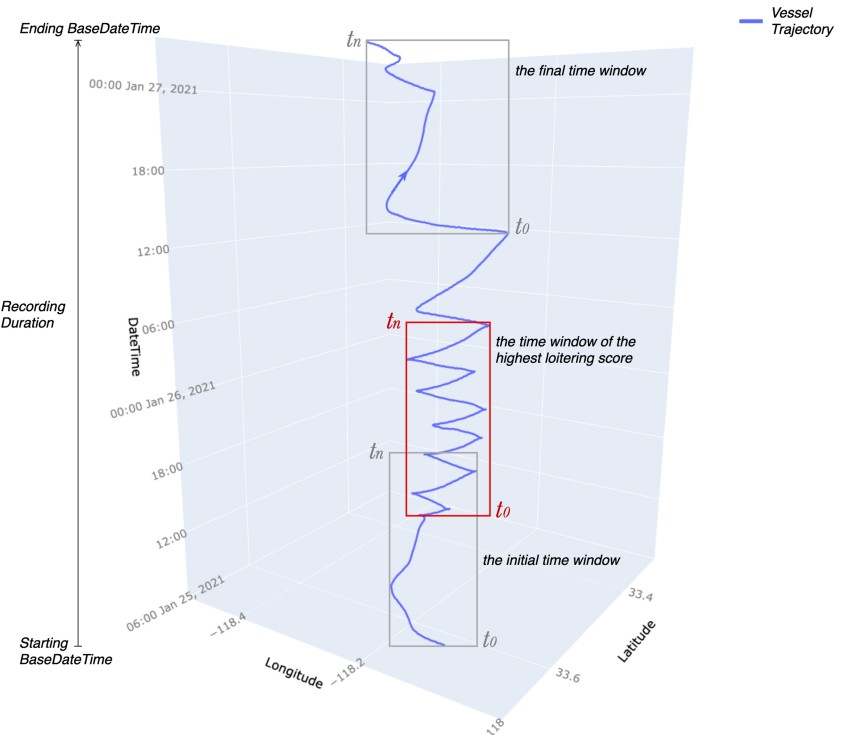

**Figure 6** The gray and red rectangles represent the time window sliding from the starting to the ending BaseDateTime of AIS messages recording duration. $t_n - t_0$ is the length of the time window which is equivalent to the length of the trajectory segment within it. The red rectangle illustrates the time window of the trajectory segment with the highest loitering score.

As described in the original paper (*Liu, Ting & Zhou, 2008*), the Isolation Forest algorithm returns a score of anomaly for each instance. The anomaly score $s$ of an instance $x$ is derived from its path length $h(x)$, which is measured by the number of edges $x$ traverses an iTree from the root node until the traversal is terminated at an external node. Based on the score $s$, the assessment is made as follow: (1) if instances return $s$ very close to 1, then they are definitely anomalies, (2) if instances have $s$ much smaller than 0.5, then they are to be considered as normal instances, and (3) if all instances return $scores \approx 0.5$, then there is no noticeable anomaly in the dataset. Adopting the algorithm, trajectories with score very close to 1 are considered as loitering, while those of score much smaller than 0.5 are regarded as normal. If all scores are nearly 0.5, then the processed set of trajectories are of similar characteristics, which means loitering instance is unlikely to exist. The scores are sorted in descending order resulting in a ranked list of loitering vessels. In other words, the order suggests the priority of the vessels to be handled for further investigation.

Finally, the loitering vessels' trajectories are marked and visualized on their respective geographical location along with the normal ones. This way provides an intuitive recognition and evaluation for operators at the surveillance office. The workflow of the loitering detection method of this study can be summarized as the flowchart in Fig. 7.

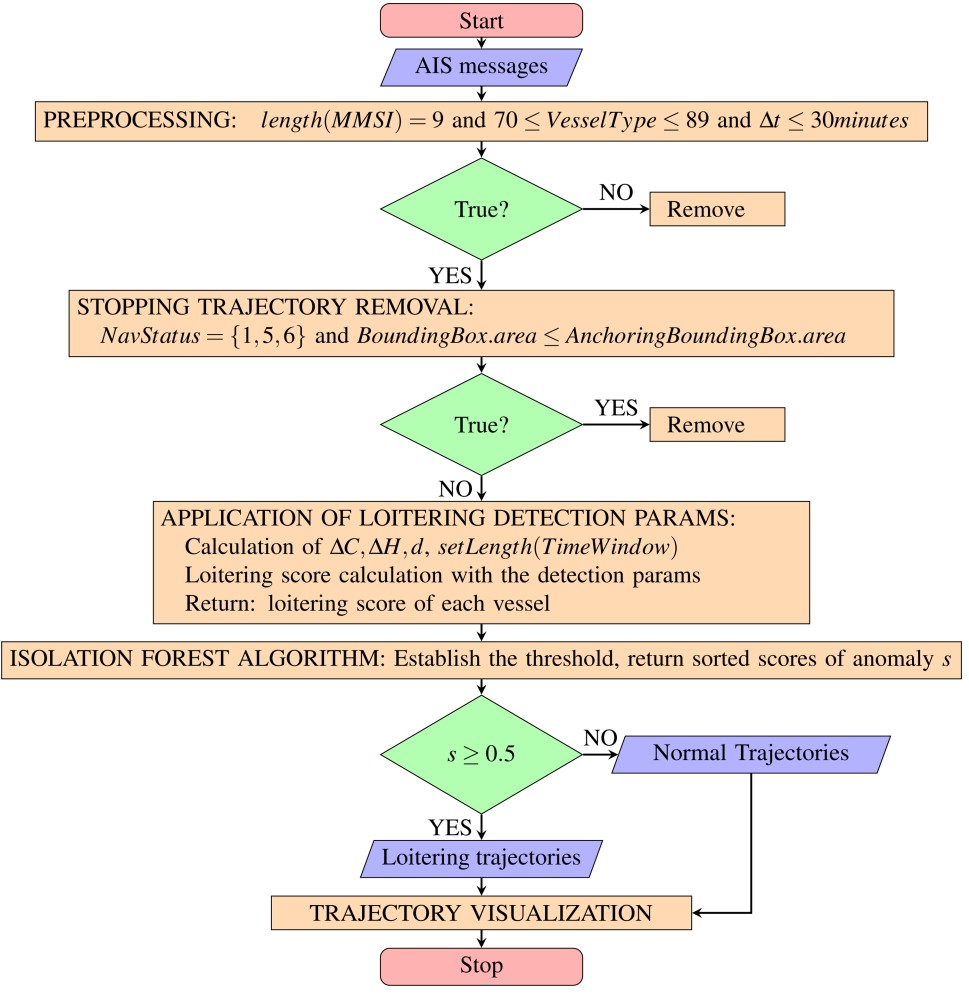

**Figure 7** **The flowchart of the loitering behavior detection framework.**

## RESULTS AND DISCUSSION

This section describes the implementation and evaluation of the proposed method based on real-world AIS data obtained from MarineCadastre. gov (*MarineCadastre, 2023*). The sea of the US West Coast is selected as the target geographic area, and a polygon is drawn to explicitly define the area as illustrated with the map in Fig. 8. The area is around 610,116.37 km². This experiment set the recording duration to 3 days. The AIS dataset is selected randomly from MarineCadastre.gov's historical AIS data of January–July 2021, and filtered with the defined polygon to precisely include only the vessels navigating through the target geographic area. After completing the preprocessing and stopping trajectory removal steps, the dataset is labeled manually and verified by an expert of the Indonesian Navy's Maritime Intelligence Office. It consists of 112 normal and 25 loitering (anomalous) vessels.

As explained in the previous section, the removal of anchoring vessel tracks is also conducted by manipulating the relevant historical AIS data to get the largest bounding

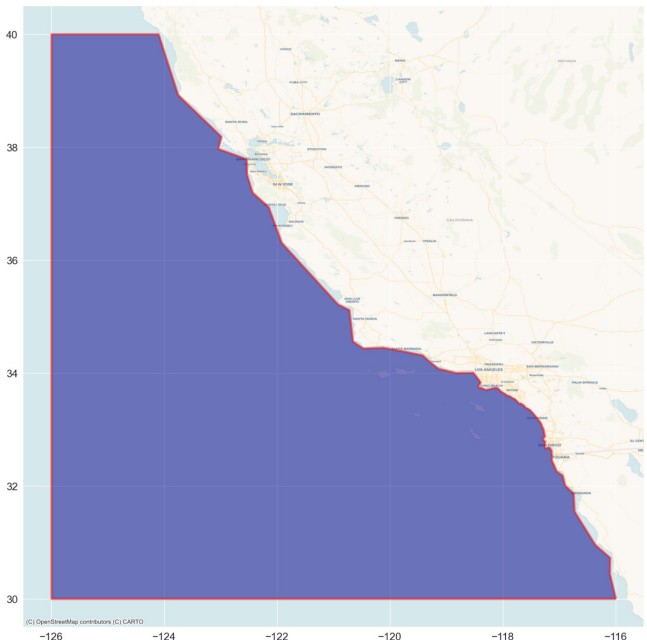

**Figure 8** The dark blue shaded area enclosed by a red line is the target geographic area. Basemap © CARTO © OpenMapTiles © OpenStreetMap contributors.

box amongst the tracks of vessels at anchor. This bounding box is used as the threshold to remove all tracks of anchoring vessels. The calculation is done as follows. First, retrieve all anchoring vessel tracks of types cargo and tanker from the historical AIS data of 1st January 2021 by applying *NavStatus* = 1 filter. This returns 353 vessels. Second, since the movement of an anchoring ship resembles circular motion as illustrated in Fig. 9A, the track bounding box should be square. The radius $r$ of the circle can be obtained by $r = \frac{\sqrt{A}}{2}$, where $A$ is the area of the square. Figure 9B depicts the distribution of the circular motion radius of the anchoring vessels. The modified Z-score method is applied to remove the outliers. Third, having removed the outliers, the largest radius of the remaining instances is 418.32 yards, which means the largest area of anchoring vessel circular motion bounding box is $699,966.49 \approx 700,000$ yards². Any vessel trajectories whose area of the bounding box $B \leq 700,000$ yards² are assumed as anchoring vessels.

In the application of the loitering detection parameters steps, this experiment employs loitering detection parameters of both Eqs. (6) and (7). These parameters are implemented in two scenarios: separated and integrated implementations. In the separated implementation, each loitering score returned from the parameters is fed to the Isolation Forest algorithm to calculate the anomaly score and distinguish loitering from normal trajectories. The scores returned by the algorithm are directly used to rank the loitering vessels to determine their priority for further assessment. While, in the integrated case, the anomaly scores returned by the Isolation Forest algorithm of both parameters are integrated to return only one anomaly score. Let $s_{f_1}$ and $s_{f_2}$ be the scores of anomaly of the parameters $F(c)$ and $F(c, h, d)$ respectively returned by the Isolation Forest algorithm,

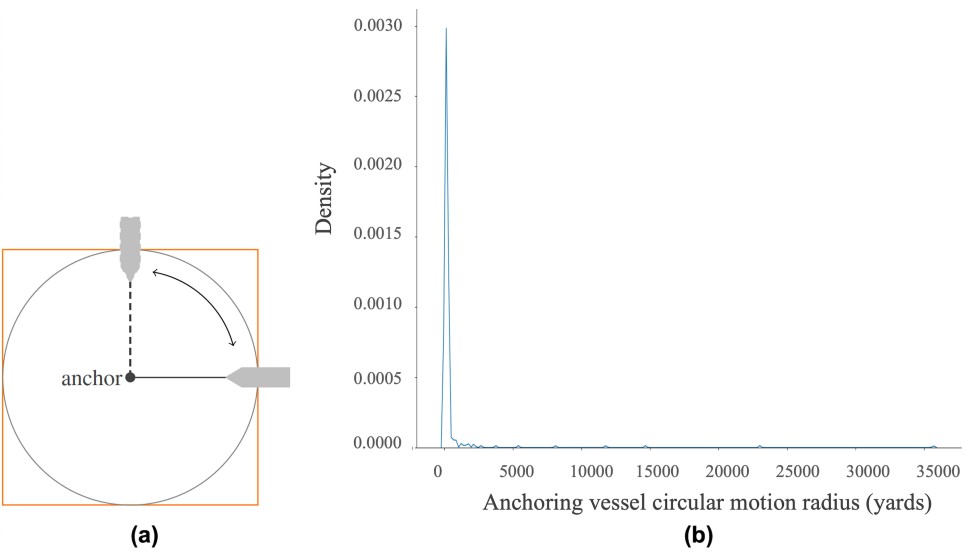

**(a)**          **(b)**

**Figure 9** **Anchoring vessel circular motion with its square bounding box (A) and the distribution of the circular motion radius of 353 vessels at anchor spread over the West and East coast of the USA (B).** The outliers are obviously observable.

and $w_{f_1}$ be the weight of $F(c)$ while $w_{f_2}$ is of $F(c, h, d)$, then the integrated anomaly score $I(f_1, f_2)$ of the parameters $F(c)$ and $F(c, h, d)$ is obtained with the formula as Eq. (11). In this case, the Entropy Weight Method (EWM) (*Zhu, Tian & Yan, 2020*) is adopted to set the weight of each parameter. A parameter with higher entropy, which means a higher differentiation degree, is given the heavier weight.

$$I(f_1, f_2) = \frac{s_{f_1} \times w_{f_1} + s_{f_2} \times w_{f_2}}{w_{f_1} + w_{f_2}} \tag{11}$$

To measure the effectiveness of the proposed detection parameters, the detection parameters proposed by *Zhang et al. (2022)* a.k.a. Trajectory Redundancy (TR) as in Eq. (1) is implemented as a benchmark. In addition, the extended version of TR as Eq. (8) proposed in this study is also applied to verify the validity of this research's hypothesis that the detection performance of TR can be improved by involving speed and COG. Table 2 describes the experiment results of all implemented loitering detection parameters. It is evident that the detection performance of the TR can be improved by extending the parameter to include speed and COG features in addition to the length of trajectory. Meanwhile, both of the proposed parameters in this study, $F(c)$ and $F(c, h, d)$, achieved significantly better results than both the TR and its extended version in all metrics. It means that these parameters are able to comprehensively represent the features of loitering behavior. Interestingly, the metrics show that the $F(c)$ performs better than the $F(c, h, d)$ even though it employs fewer features. The reason behind this outcome is the utilization of the degree of the course change $\frac{\Delta C}{180}$ in $F(c)$ calculation. The integration of $F(c)$ and $F(c, h, d)$ represents the utilization of all spatiotemporal characteristics of loitering behavior. Thus, it outperforms the implementation of $F(c)$ in almost every metric except for the specificity.

**Table 2** The metrics of all detection parameters to detect loitering trajectories (anomaly).

| Detection params | Accuracy | Specificity | Precision[*] | F-Score | Undetected anomalies | False negative |
|---|---|---|---|---|---|---|
| TR | 0.87 | 0.84 | 0.60 | 0.70 | 4 | 14 |
| $TR_E$[**] | 0.92 | 0.92 | 0.72 | 0.81 | 2 | 9 |
| $F(c)$ | 0.95 | 1.00 | 0.78 | 0.88 | 0 | 7 |
| $F(c, h, d)$ | 0.93 | 0.96 | 0.75 | 0.84 | 1 | 8 |
| Integrated[***] | 0.97 | 0.96 | 0.89 | 0.92 | 1 | 3 |

Notes.
[*] Negative precision.
[**] The extended version of TR.
[***] $F(c)$ and $F(c, h, d)$ weighted integration.

It achieved 97% accuracy and 92% F-score with 1 undetected anomaly and three false alarms, whereas $F(c)$ is able to detect all anomalies but has seven false alarms with 95% accuracy and 88% F-score. In real world scenario, it might be more reasonable to have the capability of detecting all anomalies (specificity = 100%) with the drawback of having more false alarms than having fewer false alarms but let some anomalies undetected.

The final step of the experiment is to present the ranked list of loitering vessels, and the visualization of the detection result on the pertinent geographic area. Table 3 shows the list of loitering vessels sorted with rank, while Fig. 10 illustrates the ranked loitering trajectories of which the darkest red indicates the highest priority followed by the lighter reds of the lower, and the lightest is the lowest. Furthermore, Figs. 11 and 12 depicts the actual and prediction of the loitering vessels trajectories amongst that of the normal movements and compares the prediction results of the TR parameter with the weighted integration of $F(c)$ and $F(c, h, d)$ parameters.

## CONCLUSIONS

The abundant availability of AIS data stemming from the widespread application of shipborne AIS has facilitated researchers to implement computational methods in analyzing vessels' behavior and propose solutions for anomaly detection in maritime traffic. However, this study found that the existing anomaly detection approaches tend to be constrained by the geographic regions where the model is trained and the requirement to construct a profile of normal instances to identify anomalies. Above all, ship loitering behavior is the least explored abnormality in the maritime domain even though it is the most observed anomaly in the real world scenario, especially for vessels of types cargo and tanker. Based on the quantification of loitering behavior characteristics with the dynamic information of AIS messages, this article proposed parameters to spot loitering trajectories. The experimental results on a real-world dataset prove the efficiency and effectiveness of the proposed method. The effectiveness is confirmed by all of the evaluation metrics. It evidently outperforms the existing approach with remarkable accuracy and an F-score of 97% and 92%, respectively. The efficiency is demonstrated by the versatility of the method to be applied in a nationwide area without the burden of training normal instances to build models of normalcy. The experiment produces a priority list of loitering vessels ranked with

**Table 3  List of loitering vessels ranked with the score of anomaly.**

| No | Vessel MMSI | Integrated score of anomaly |
|---|---|---|
| 1 | 373932000 | 0.88 |
| 2 | 538005018 | 0.79 |
| 3 | 357051000 | 0.75 |
| 4 | 372399000 | 0.73 |
| 5 | 477969700 | 0.72 |
| 6 | 371443000 | 0.72 |
| 7 | 565747000 | 0.71 |
| 8 | 477848700 | 0.71 |
| 9 | 431496000 | 0.70 |
| 10 | 303352000 | 0.70 |
| 11 | 538005965 | 0.69 |
| 12 | 538005057 | 0.68 |
| 13 | 249414000 | 0.67 |
| 14 | 352413000 | 0.66 |
| 15 | 220593000 | 0.65 |
| 16 | 256937000 | 0.64 |
| 17 | 211335760 | 0.64 |
| 18 | 211327410 | 0.62 |
| 19 | 538008361 | 0.60 |
| 20 | 338500000 | 0.59 |
| 21 | 356581000 | 0.59 |
| 22 | 366576000 | 0.57 |
| 23 | 636017857 | 0.56 |
| 24 | 538007675 | 0.56 |
| 25 | 351249000 | 0.53 |
| 26 | 367067110 | 0.52 |
| 27 | 636020456 | 0.50 |

the scores of anomaly as well as an intuitive visualization in the relevant geographic area. In the real-world scenario, the priority list and the geographic visualization are practical means to facilitate further anomaly investigation by human operators. The future development of this study is to perform an even deeper analysis of the loitering trajectory features with the integration of the static and voyage-specific information of the AIS messages to uncover the activities behind the behavior. It is to answer the question of which loitering trajectories are related to safety, security, or other issues that are not observable by the conventional surveillance system.

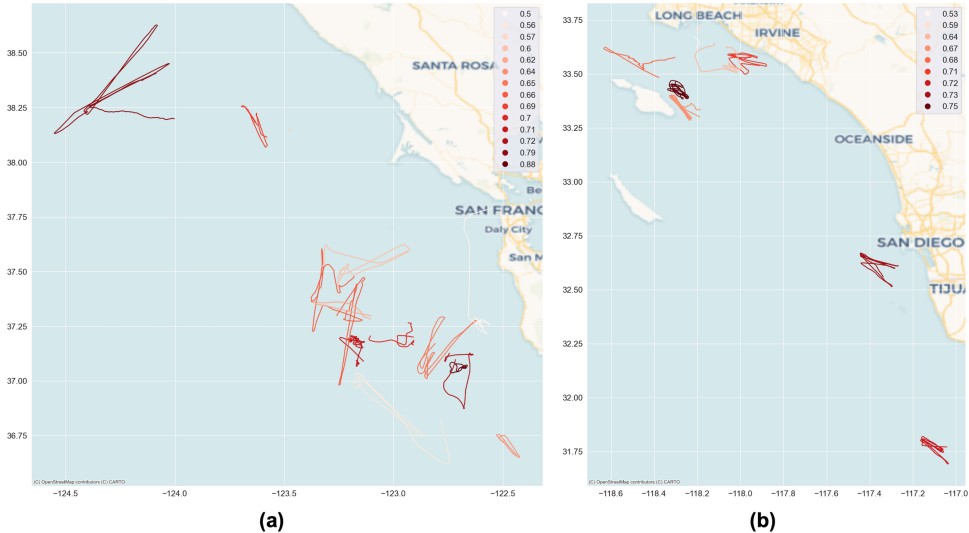

**Figure 10** **Visualization of the loitering trajectories ranked with the score of anomaly.** The darkest red indicates the highest score, while the lightest is the lowest one. (A) The Northern half; (B) the Southern half area. Basemap Ⓒ CARTO Ⓒ OpenMapTiles Ⓒ OpenStreetMap contributors.

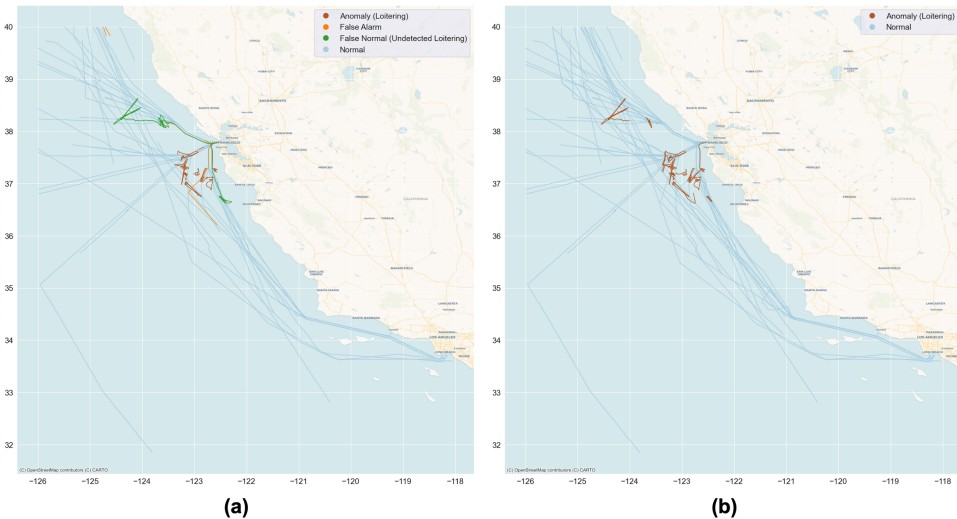

**Figure 11** **Prediction results visualization of the Northern half of the target geographic area.** The brown lines are predictions of loitering trajectories that matched the actual loitering, while the light blue lines are predictions of normal that matched the actual normal. The orange and green lines are the false alarms and undetected loitering respectively. (A) The prediction of the TR parameter with 87% accuracy; (B) the weighted integration of the $F(c)$ and $F(c, h, d)$ parameters that improved accuracy to 97%. Basemap Ⓒ CARTO Ⓒ OpenMapTiles Ⓒ OpenStreetMap contributors.

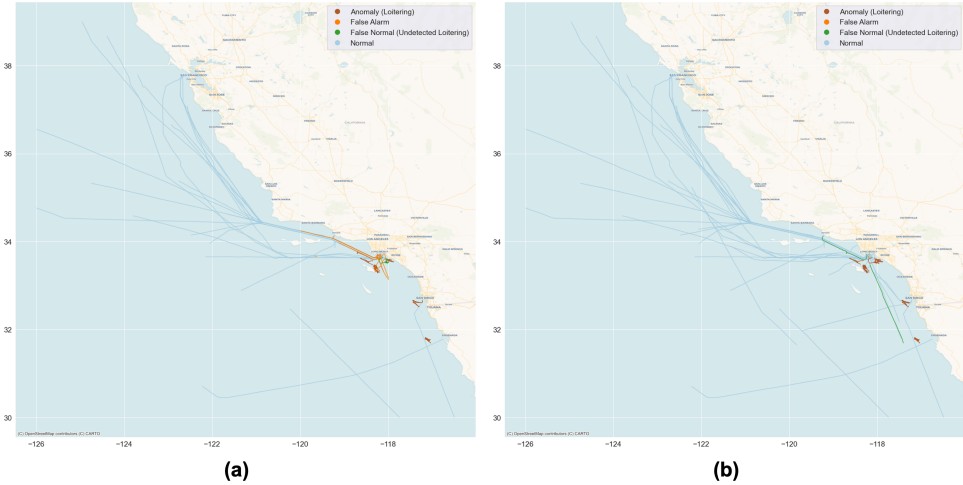

**Figure 12** **Prediction results visualization of the Southern half of the target area.** The brown lines are predictions of loitering that matched the actual loitering trajectories, while the light blue lines are predictions of normal that matched the actual normal. The orange and green lines are the false alarms and undetected loitering respectively. (A) The prediction of the TR parameter with 87% accuracy; (B) the weighted integration of the $F(c)$ and $F(c,h,d)$ parameters that improved accuracy to 97%. Basemap © CARTO © OpenMapTiles © OpenStreetMap contributors.

## Funding
The authors received no funding for this work.

## Competing Interests
The authors declare there are no competing interests.

## Author Contributions
- Wayan Mahardhika Wijaya conceived and designed the experiments, performed the experiments, analyzed the data, performed the computation work, prepared figures and/or tables, authored or reviewed drafts of the article, and approved the final draft.
- Yasuhiro Nakamura analyzed the data, authored or reviewed drafts of the article, and approved the final draft.

## Data Availability
The codes for loitering score calculation with F(c) and F(c,h,d), and calculating the scores of anomaly and the evaluation metrics of each prediction, are available in the Supplementary Files.

## Supplemental Information
Supplemental information for this article can be found online at http://dx.doi.org/10.7717/peerj-cs.1572#supplemental-information.

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
