# Peer review of "Loitering behavior detection by spatiotemporal characteristics quantification based on the dynamic features of Automatic Identification System (AIS) messages"

_PeerJ Computer Science, doi:10.7717/peerj-cs.1572_

## Round 0.1 · original submission · Minor Revisions

Please pay careful attention to addressing the reviewers' comments in your revisions. The reviewers raised a range of important issues, but also issues that I believe should be possible to address.

In addition to those comments, I also had two comments I wanted to add to those of the reviewers:
1. Like both reviewers, I found the use of the term "anomalous" in the paper confusing. I believe this is because the term is used in multiple different senses in the paper in different contexts. Please take particular care to address this ambiguity in revision and help the reader more easily understand the differences. I believe several of the reviewers' comments will become much easier to answer once you have clearly distinguished apart the different senses in which the term is used (including, for example, the sense of "true positive loitering behaviour" and the subtly different sense of "output from the Isolation Forest algorithm").
2. In addition, I would have liked to see more information about the manual labelling of loitering. Who performed this? What quality controls were applied to the manual labelling process? I would also like to see the data set of manually labeled vessels as a separate supplemental file to the paper. (Note: "Methods should be described with sufficient information to be reproducible by another investigator.")

Reviewer 1 ·

Basic reporting

The raws have been shared and are of good, professional quality.

The abstract should be clearer on why anomaly detection is important. A specific type, loitering, has been singled out in this paper as it apparently occurs frequently and has not been studied well. For context, the other common forms of anomaly should be discussed.

Lines 92-93 make a statement without providing a citation or further explanation or examples: “Meanwhile, in the dynamic of ship behaviour in real-world situations, anomaly and normal events often overlap with each other.”

The introduction provides a preview of the proposed method from line 95 following. The description is overly convoluted and should be simplified.

Lines 211-212 discuss some limitations of previous work but it is not clear what the impact of these limitations is.

Lines 224-225 make a statement without providing representative citations.

The literature review provides several different definitions of abnormal behaviour at sea and loitering in particular. It would be beneficial to tell a clearer story of the different categories of efforts in this space. The literature review appears to lack a clear structure. At the same time the presented figures are very appropriate for illustrating the main challenges in this space and are very effective and clear.

Experimental design

The proposed article suggests original primary research within the aims and scope of the journal.

The research question has very practical application and it is indeed difficult to find work specifically on the topic of loitering. I believe that the problem is receiving more attention recently but the research body on this is still sparse. The approach is somewhat against the current trend of applying a deep learning technique to everything is instead proposes a different methodology that might be independent of region specific characteristics.

The definition of loitering is extended in lines 269-270 and it should be clarified why this is necessary and the impact the change of definition has. The is implied by the following description of the experiment, but it should be specifically mentioned for clarity.

The research methods are described with sufficient detail and could be replicated.

Validity of the findings

All underlying data have been provided

The paper considers a specific type of vessels, namely those fitted with an AIS. It would be beneficial to clarify what percentage of general ocean traffic this applies to as this also indicates the importance of this work.

It would be interesting to hear how this proposed method performs in comparison to existing methods or the current deep learning trend. What is the advantage of this approach?

The conclusions are well stated and linked to the original question.

Additional comments

n/a

Reviewer 2 ·

Basic reporting

The paper is clearly written and easy to understand. Scenarios that existing method do not work and directions for improvement are presented in the literature review which makes transaction from the existing method to your new approach easy to follow. However, it requires more sign posting to make it really clear at the beginning distinguishing your edits from the original TR (trajectory redundancy).

Section references are missing (e.g., line 115)

Figure 6 is ambiguous as it is not quite clear what x axis and y axis mean: the trajectories seem to suggest that x and y are spatial axes while the the time windows suggest the y direction represents time.

Experimental design

In general, the paper focuses on a problem of anomaly detection based on AIS trajectory data. The investigation process is sound and recorded in detail.
One decision that does not quite make sense to me is the decision to only looking at tanker and cargo ship despite the explanation given from line 255 to line 261. Since loitering is quite a general mobility behaviour, it should not be vessel type dependent. Meaning a trajectory that generated by a tanker that classified as loitering should also be classified as loitering if it were generated by another type of vessel, although the level of anomalousness could be vessel type dependent. I would advise the authors to precisely define terms like loitering and anomalous behavior, and the relations between three aspects of a trajectory 1. whether it is loitering; 2. whether it is anomalous; 3. the type of the vessel.

Validity of the findings

The 0.5 threshold chosen for classification based on anomaly score given by isolation forest seems quite opportunistic. The value should be dataset/vessel type dependent as some dataset may contain more anomaly behavior than the others. It would make more sense to employ a mechanism to learn that threshold from data. So I would suggest introducing more experiments to either demonstrating the 0.5 threshold works for other datasets as well or introducing a learning process to set this threshold (this however may undermine the proposing algorithm as it is stated to be training free)

---

## Round 0.2 · accepted · Accept

Thank you for the careful revisions. I'm comfortable that you have adequately addressed all the issues raised..